# Evaluation of Meat Safety Practices and Hygiene among Different Butcheries and Supermarkets in Vhembe District, Limpopo Province, South Africa

**DOI:** 10.3390/ijerph20032230

**Published:** 2023-01-26

**Authors:** Bridget Jabulile Siluma, Ephraim Tsietsi Kgatla, Bono Nethathe, Shonisani Eugenia Ramashia

**Affiliations:** 1Department of Food Science and Technology, Faculty of Science, Engineering, and Agriculture, University of Venda, Thohoyandou 0950, South Africa; 2Faculty of Applied Sciences & Biotechnology, School of Bioengineering & Food Technology, Shoolini University India, Bajhol 173229, India

**Keywords:** meat safety practices, hygiene, butcheries, supermarkets, South Africa

## Abstract

Good hygienic practices are required to reduce the risk of microbial contamination during meat processing. We evaluated good hygiene and meat safety practices among different village butcheries (6), commercial butcheries (8), and supermarkets (18) through direct personal observations. The supermarkets and commercial butcheries wore personal protective equipment (PPE) and used proper waste procedures. Moreover, there were pest control devices, a safe water supply, and staff handling money away from meat. At village butcheries, wearing hairnets and aprons, and the display of raw meat being separate from offal were identified as good practices. The irregular washing of hands (67%), less use of gloves (83%), wearing of open sandals (67%) and jewelry (33%), use of the same coat for different activities (100%), lack of paper towels (100%) and pest control devices (67%) and mismanagement of waste (33%) were practices that led to unsafe meat handling. Our study identified good meat safety practices at supermarkets. A combination of good and unhygienic meat handling practices were identified at commercial and village butcheries. These findings suggest a need for intervention through training on food safety in order to improve the hygienic practices of meat handling along the beef supply chain, more especially in commercial and village butcheries.

## 1. Introduction

The increased demand for foods of animal origin is often linked to the world’s growing human population [1]. Consequently, meat producers, processors, and consumers give higher importance to meat safety [2]. The main source of protein, vitamins, and nutrients for the development and functioning of body cells is meat [3]. Worldwide, foodborne diseases are associated with the consumption of spoiled foods, which may occur during processing, among which meat processing has been attributed as a primary source of illness when contaminated [4]. Foodborne illnesses are prevalent in developing countries due to poor food handling and sanitation practices, insufficient laws for food hygiene, weak regulatory systems, lack of funding for the purchase of the necessary equipment, and a lack of food-handler education [5,6].

The main source of foodborne diseases is through ingestion of meat contaminated by pathogenic bacteria such as *Staphylococcus aureus*, *Salmonella* species, *Listeria monocytogenes*, *Escherichia coli 0157:H7* and *Campylobacter* species [7]. Meat that is improperly handled may result in meat contaminated by pathogenic bacteria and can lead to health hazards for the consumer [8]. Butcheries have a massive role in the prevention of meat-borne diseases because of the high chances of meat contamination at the butchery level. Practice and maintenance of proper hygiene during meat handling is necessary for the provision of healthy and fresh meat for human consumption [9].

Often, meat handlers’ poor personal hygiene operates as a vector for the spread of microbes through their hands, wounds, lips, skins, and hair [10,11]. If proper sanitation and hygiene procedures, such as washing hands, wearing protective clothes, cleaning and sanitizing butchery equipment and utensils, are not followed, bacterial contamination, meat loss, and post-harvest meat shortages arise [12]. According to the findings of the study conducted in Tanzania by Ntanga et al., 2014 [13] and Birhanu et al., 2017 [14] in Ethiopia, the bacterial load in the meat, meat contact surfaces and utensils from the butcheries taken through swabs was higher than what was considered acceptable.

The wholesomeness of meat is a shared responsibility for all individuals in the food chain. To correct the errors from farm to fork, there is a deep need of education and training in the prevention of foodborne diseases among abattoir workers, butchery, meat producers, suppliers, handlers, and the general public [11]. Standard and hygienic ways of handling and processing meats are generally neglected in developing countries [15]. According to the World health Organization, foodborne illnesses are estimated to have caused 600 million cases, 420,000 deaths, and approximately 33 million years of life of impairment worldwide in 2010, with Africa facing the greatest burden of mortality [1,16]. In order to reduce microbial contamination, hygienic handling techniques during preparation, distribution, storage, and retail sales must be improved [1]. For health and safety reasons, it is essential to always wear protective gear and wash hands before and after selling meat [13]. Wearing of an apron or gown during meat handling is an important practice that aims to protect both the meat handler and the meat from exposure to foodborne pathogens [17].

Several studies investigated meat safety knowledge and practices [8,9,18], while others determined the handling of meat practices along the beef supply chain [12,17] and bacteriological quality of meat from abattoir and butcher shops [17,19] in different countries. There is a critical need in the literature to investigate the practices of food handlers in their everyday activities of employment and the potential sources of microbiological contaminants that can impair the quality of meat products [2]. When it comes to bacterial illnesses that spread through the consumption of meat and meat products, there is little information available about the precise amount of exposure of different populations to potential dangers [5].

The presence of hygiene measures has an impact on hygiene, however, developed countries with excellent levels of hygiene also have foodborne illnesses [20]. In South Africa, studies on meat safety practices and hygiene were done among slaughterhouse workers [6], as well as studies on game meat production for animal class and health compliance [21], on the management of meat safety in abattoirs [22] and on the traditional slaughter of goats [23]. To protect the population from food-borne bacterial diseases, it is necessary to educate and campaign for proper sanitation and meat-handling practices in abattoirs and butcher shops [1,8,24]. No documentation was available with regards to meat safety practices and hygiene among butcheries and supermarkets in Vhembe district, Limpopo province, South Africa. Thus, the objective of this study was to evaluate meat safety practices and hygiene among different butcheries and retail supermarkets in Vhembe district. The results of this study may provide information on whether good manufacturing practices of meat are being fully followed at the retail level and whether they pose a threat to the health of the public.

## 2. Materials and Methods

### 2.1. Study Settings

The study was conducted from October to November 2021 at thirty-two butcheries and supermarkets found in Vhembe district, including 8 commercial butcheries situated in Thohoyandou, Shayandima, Sibasa and Elim, 6 village butcheries situated in Tsianda, Lwamondo, Matsila, Levubu, Masia and Vuwani and 18 supermarkets situated in Thohoyandou, Phangami, Sibasa, Mphephu, Biaba, Dzanani, Tshilamba and Makonde, within Vhembe district. The Vhembe district is located in Limpopo province, which is in the northern part of South Africa. Village butcheries operated as independent retail establishments and each village butchery had about 2 to 3 employed workers; commercial butcheries also operated as independent retail establishments and had about 3 to 7 employed workers on site. In supermarkets, butcheries were situated within stores that sell various products such as groceries, food supplies and baked goods. Supermarket butcheries had 5 to 8 workers on-site at the time of the study.

### 2.2. Study Design and Data Collection

Data were collected through direct personal observation using a structured questionnaire survey checklist to assess beef meat safety and hygiene practice in various village butcheries, commercial butcheries, and supermarkets (Table 1). The survey checklist was adapted and modified from questionnaires and survey checklists in similar previous studies [1,6,24,25]. The questions involved the following themes: (i) socio-demographic characteristics of the participants; (ii) hygiene of meat handlers; (iii) cleanliness of working clothes; (iv) infrastructure, and maintenance of hygiene in supermarket/butchery; (v) the display of meat in butchery/supermarket. The survey checklist was administered for each retail shop (*n* = 32) being assessed and was based on the South African regulations of the Department of Health on the General Hygiene Requirements for Food Premises, Food Transport, and Related Matters R 638 (Act 54 of 1972): Food stuffs, cosmetics & disinfectants Act, 1972 (Act 54 of 1972)

### 2.3. Data Management and Analysis

The collected data were entered into a Microsoft Excel spread sheet (Microsoft, Redmond, WA, USA) and analyzed using IBM SPSS statistics for Windows, version 28 (IBM Corp., Armonk, NY, USA). The data were summarized using descriptive statistics, including frequency and percentage. A one way analysis of variance (ANOVA) was used to assess the difference between the various butcheries. A *p*-value of less than 0.05 was set as a significance level. The meat safety practices at various butcheries were described descriptively.

## 3. Results

### 3.1. Sociodemographic Characteristics of Meat Handlers

Table 2 summarizes the sociodemographic characteristics of the butchery workers from various butcheries in the Vhembe district. Workers (*n* = 177) were both males and females ranging from 18 to 54 years. Most (68%) participants at the supermarket butcheries studied up to secondary school and only a few (6%) from the village butcheries obtained primary education. Most of the butchery workers earned between 3000 to 5000 Rand per month, which means that they belonged to the poor and lower middle class income brackets in South Africa.

### 3.2. Meat Safety Practises and Hygiene of Meat Handlers at Butcheries

The majority (72%) of supermarket butcheries required workers to wash hands prior to work and many (89%) used protective gloves before handling meat. All supermarket butcheries workers wore personal protective equipment such as an apron or coat, protective boots and a hairnet while handling meat. Among the retail shops, 67% wore long protective clothes. However, they did not completely cover personal clothes. Seventeen percent of the retailers had staff workers wearing jewelry (watch, bracelet, and ring) while handling meat.

Among the commercial butcheries, 50% washed their hands. The majority (75%) wore gloves prior to meat handling, while 25% handled meat with either bare hands or a plastic bag before distributing it to the consumer. Aprons or coats were worn by all commercial butchery workers; while 62% wore the same protective clothes when carrying out other activities in the butchery throughout the day, 75% wore hairnets. In all commercial butcheries, staff workers handling money were separate from those handling meat. Wearing of jewelry was not observed in 87% of the commercial butcheries.

At the village butcheries, 67% washed their hands prior to meat handling, however this was with water placed in a plastic basin/bucket. The water utilized had been previously used to wash hands and contained a dish cloth used to wipe the hands and counters of the butchery. Most of the time, the water contained dishwashing liquid or soap. Protective gloves were worn in only 17% of the village butcheries most of the butcheries (83%) handled meat with bare hands or a plastic bag. All village butchery staff wore either an apron or gown; 33% wore protective boots, 67% wore open sandals or shoes. All village butchery workers wore the apron or coats while carrying out other activities besides meat handling. At the village butcheries, all workers handling money were also handling meat. In Approximately 33% of the village butcheries, workers were observed to be wearing jewelry. Table 3 summarizes the observational assessments on meat safety practices and hygiene at supermarkets, as well as commercial and village butcheries.

### 3.3. Hygiene of Working Clothes at Various Butcheries

The majority (67%) of supermarket butchery workers had recent dirt (fresh particles of meat or blood) on work clothes and 17% of butcheries had ingrained dirt (old particles of meat and blood stains) on work clothes. At commercial butcheries, 38% of the butcheries had recent dirt covering working clothes, and 22% had ingrained dirt on their work clothes. Many (100%) of the village butcheries had recent dirt covering their work clothes and 33% had ingrained dirt on them.

### 3.4. Infrastructure and Maintenance of Hygiene at Butcheries

All supermarket butcheries had walls, floors, and ceilings in good condition. All the retail shops had a safe water supply and pest control devices. However, 61% had dirty floors. Fifty-eight percent had clean counter and hooks; moreover, seventy-two percent had clean cutting tables for meat and a waste management system. Among the commercial butcheries, 87% had good structures. however, 13% had cracked walls and floors and 38% contained dirty floors. Sixty-two percent were identified with cleaning cloths and detergents and seventy-five percent of them had pest control devices. None of the commercial butcheries utilized paper towels. However, 75% percent of them properly confined and disposed waste. At the village butcheries, 50% of the structures had walls, floors, and ceilings in good condition, while 50% had ceilings and walls that were tearing down and where we observed cracked tiles on the floor. Seventy-five percent of the village butcheries had a safe water supply, while twenty-five percent had water supplied from water reservoir tanks instead of directly from the tap. The majority of the village butcheries (83%) confined, as well as properly managed and disposed of waste. However, 17% lacked a dustbin and disposed waste on the dumpster site outside of the building of the butchery. Approximately 67% of the village butcheries had cleaning cloths, detergents, and pest control devices.

### 3.5. Display of Meat at Butcheries

Among the supermarket butcheries, 83% displayed meat of different species separate from offal on a meat display fridge. In 89% of the supermarket butcheries, meat appeared red in color without an unpleasant odor and in 17% of the supermarket butcheries meat appeared dark brown. The majority (87%) of the commercial butcheries displayed meats of different species separate from offal in a window display fridge, 62% had meat that appeared red in color and 38% had meat that appeared dark brown and had an unpleasant odor. At the village butcheries, 67% displayed meats of different species and offal separately in a window display, 83% of the butcheries had meat that appeared red in color, while 17% had meat that appeared dark brown with unpleasant odor. The results for the display of meat in various butcheries is summarized in Table 3.

### 3.6. Statistical Analysis of Variance of Meat Safety Practices among Various Butcheries and Supermarkets

Table 4 summarizes the mean and standard error of meat safety practices among various butcheries and supermarkets in the Vhembe district., Limpopo, South Africa. The village, commercial and supermarket butcheries did not significantly differ based on the washing of hands, wearing of apron, gloves, hairnets, jewelry, butchery shop floor cleanliness, cutting tables, water supply, detergents, and pest control devices (ANOVA *p* > 0.05). However, there was a significant difference between the various butcheries in different locations with respect to the use of waterproof boots (*p* = 0.01), disposable paper towel availability (*p* = 0.04), waste management (*p* = 0.025), the condition of the structure of the butchery (*p* = 0.05), staff preparing raw meats being separated from ready-to-eat meats (*p* = 0.03), staff handling money being separated from handling meat (*p* = 0.00) and meat appearing red/brown and having unpleasant odor (*p* = 0.029).

Employees at village butcheries washed hands more than those at commercial butcheries (Mean = 0.67, standard error (SE = 0.211)), however workers at commercial butcheries wore gloves, hairnets, and waterproof boots more than village butcheries (Mean = 0.75, standard error (SE = 0.164)); all butcheries had a water supply, however, some village butcheries had water supplied from reservoir tanks instead of directly from the tap. This water had been collected from the nearest borehole, transferred into the reservoir tanks, and stored for present and future use. Moreover, they wore aprons while handling meat (Mean = 1.00, standard error (SE = 0.00)).

## 4. Discussion

Due to the high likelihood of meat contamination at the butchery level, butcheries play a significant part in the prevention of meat-borne diseases. For the purpose of providing safe and fresh meat for human consumption, it is vital to practice and maintain good hygiene during meat handling [9]. We evaluated meat safety practices and hygiene among supermarkets, commercial and village butcheries in the Vhembe district, Limpopo, South Africa. This study was motivated by the need of information to guide food safety policy development, good manufacturing practices and training in meat handling and hygiene in butcheries of all levels. The discussion that follows focuses on the primary meat processing techniques and their potential public health implications. Moreover, the practices are discussed considering the demands of the South African proclamations: Foodstuffs, Cosmetics & Disinfectants Act of 1972: General Hygiene Requirements for Food Premises, Food Transport, and Related Matters R 638 (Act 54 of 1972) [26]. The legislation has been designed for the protection of consumers, to secure the production and marketing of safe, nourishing, and high-quality meat and meat products, as well as to provide general guidelines for food hygiene.

### 4.1. Hygiene and Meat Safety Practices of Meat Handlers at Supermarket Butcheries

In the present study, the washing of hands, wearing protective gloves, boots, coats, and hairnets were good practices identified at the supermarket butcheries. Gutema et al. [1] also observed at slaughterhouses in Bishoftu, Ethiopia that the use of aprons, white coats, boots, and hair coverings were good practices at slaughterhouses. According to Nyamakwere et al. (2017) [6], these are important practices that protect both the meat handler and the meat from exposure to foodborne pathogens. Although some workers wore protective gloves and clothing, practicing the washing of hands before and after sales in the butchery is necessary for sanitary purposes [27] and is key to preventing the spread of infections. If not properly implemented, it can lead to disease outbreaks [28]. Therefore, the washing of hands by the majority and the wearing of protective clothes in supermarket butcheries were good meat hygiene and safety practices in line with the recommendations of the Department of Health on principles of General Hygiene Requirements for Food Premises, Food Transport, and Related Matters R 638 (Act 54 of 1972) [26], and should be encouraged.

In the present study, half (50%) of the commercial butcheries practiced washing of hands and 75% wore gloves. The present findings revealed a lack of adherence to R 638 (Act 54 of 1972) [26], which states that food cannot be handled by anyone whose hands have not been completely cleaned with soap and water or another effective cleaning method at the start of each work shift, of each day, or after a break. This, also, indicates a much lower proportion of compliant workers than that reported in the streets and open market meat vendors in Tamale Metropolis, Ghana, as reported by Adzitey et al. [29], where almost everyone (99%) of meat handlers knew that washing hands before working minimizes the risk of meat contamination. Hand hygiene is not a new principle in the food industry for preventing microbial contamination of food. Unfortunately, hand hygiene is not always practiced, nor is it practiced properly [12].

In this study, aprons, coats, or gowns were worn by all commercial butchery workers, and 75% of them wore protective boots. This indicates consistence with the Department of Health’s regulation [26] that food handling must be done while wearing protective clothes, including footwear, headwear, and other items of clothing, to prevent food from becoming contaminated. This result indicates a higher proportion of compliant workers that that reported in butcheries at Kampala district in Uganda, where Mirembe et al. [30] reported that 31.5% wore protective gear during meat handling, as well as a higher proportion that in the study conducted in Tamale Metropolis, Ghana by Adzitey et al. [29] wherein half of the meat producers (58%) did not wear an apron while working. To prevent meat-handling staff and the meat from being exposed to infections, practices such as wearing aprons, white coats, boots, and covering one’s hair should be utilized [1].

In this study, 67% of workers in village butcheries washed their hands; however, this was with water placed in a plastic basin or bucket. The water had been previously used to wash hands, with a dish cloth used to wipe the hands and counters of the butchery. The water contained dishwashing liquid or soap most of the time. This practice was not in line with the requirements of R 638 (Act 54 of 1972) [26], which states that in food establishments, there must be hand-washing facilities with hot water, if appropriate, as well as a supply of soap and clean, disposable hand-drying material or equipment, which must be available for workers to wash and dry their hands. The findings of the present study are similar to that reported along the beef supply chain in Uganda by Kyayesimira et al. [31], where in butcheries, hands were mostly washed with cold water and soap (82.6%), but there were some cases where hands were washed using a washcloth (1.4%). In Lahore, Pakistan, Mallhi et al. [32] reported that 27% of workers used a towel for hand-washing and 73% used only water for cleaning the shop, indicating a lack of control over hygiene practices. Food handlers should be equipped with the required knowledge and skills to handle food safely. Consequently, sanitation and hygiene training would be able to modify personal behavior and attitudes [29].

In this study, 17% of the village butcheries wore protective gloves. This is a concern and revealed a lack of proper education in terms of meat handling; this could also be due to a lack of finance to purchase the necessary equipment and tools to practice safe meat handling. Furthermore, this was not in consistence with R 638 [26] which states that a food handler should avoid handling unpackaged food with bare hands unless it is necessary for preparation. This report was closely similar to the report in Chitwan, Nepal by Khanal and Poudel [9], in Uganda by Jeffer et al. [33] and in Western Kenya by Cook et al. [34], in which none of the workers working in the handling of meat made use of gloves.

In this study, all village butchery workers wore either an apron or gown. This was consistent with R 638 [26], in which no one should handle food unless they are wearing the proper protective gear. This result differed from the report in Bangladesh by Banna et al. [18], where many of the meat handlers rarely used an apron (96.4%) when working. Furthermore, Jeffer et al. [33] indicated that 0% of workers in meat shops wore safety gear such as coats and gloves. According to Mbonabucha and Fweja [35], if all food workers wore protective clothes to prevent contamination of food equipment and utensils, the contamination of food could be avoided. In this study, 33% of the village butchery workers wore protective boots while 67% wore open sandals or shoes. This proportion is higher than the report in Gondar town, Ethiopia of Birhanu et al. [14] where 34% of the meat handlers wore open sandals while working, and is lower than that reported in Uganda by Jeffer et al. [33], where 0% of the workers wore gumboots while working. The Regulation R 638 (Act 54 of 1972) [26] requires that when handling meat, safety clothing, including protective boots, should always be worn; therefore, this was not consistent.

In this study, all supermarket and commercial butcheries separated staff handling money from those handling meat. According to Atlabachew and Mamo [4], this reduces possible contamination by micro-organisms and food-borne pathogens, which can result in health hazards. This situation was different at butcher shops in Gondar town, Ethiopia, where Birhanu et al. [14] reported that 45.5% of respondents handled money concurrently with serving customers. This result was consistent with the regulations of meat safety stipulated R 638 [26], in which a person must not handle food after handling money. Cash is commonly used to exchange products and services in countries all around the world and is exchanged often in many types of business and can provide a vast surface area for pathogens to thrive. Therefore, the separation of workers handling money from those that handle meat is a good meat safety practice and should be encouraged [7].

In this study, at all village butcheries, staff handling meat were also handling money. This result is closely similar to the report by Kyayesimira et al. [31] in Uganda, in which most meat handlers (93.5%) also worked as cashiers, thereby handling both money and meat at the same time. This is also consistent with the report of Mallhi et al. [32] in Lahore, Pakistan, where 71.8% of butchers handled money with their bare hands when handling and cutting meat cuts; this practice could act as a pathogen carrier. During retailing or serving, the individual handling money should not be allowed to handle meat. This is because money is unclean and can contaminate food handler’s hands. According to a study by Simon-oke and Ajileye in Akure, Nigeria, parasites such as *Enterobius vermicularis*, hookworm, *Giardia lamblia*, *Ascaris lumbricoides*, *Hymenolepis nana*, *Strongyloides stercoralis*, *Trichuris trichiura*, *Isospora belli*, *Entamoeba histolytica*, *Balantidium coli* and flagellates were recovered on currency notes sourced from food vendors and butchers [36].

Ascariasis is a condition brought on by *A. lumbricoides*. Abdominal pain or intestinal obstruction may result from an Ascaris worm infection. Amoebiasis caused by *E. histolytica* is transmitted orally by ingesting the cyst. Amoebic dysentery, liver abscess, and even mortality can be consequences of amoebiasis [36]. Furthermore, in a study conducted in Nairobi, Kenya, by Kuria et al., 2009, pathogenic bacteria including *E. coli*, *Klebsiella*, *Serratia*, *Enterobacter*, *Salmonella*, *Acinetobacter*, *Enterococci*, *Staphylococcus* and *Bacillus cereus* were discovered in coin samples obtained from taxi drivers, butchers, food restaurant attendants, grocery store attendants, roast maize sellers, and students [37].

Money has been handled by a wide range of people, including butchers, who may have contaminated the notes with blood. Blood serves as a good medium for bacterial growth. Moreover, other possible handlers of the notes such as traders, beggars, and people who conduct other jobs could contaminate hands and result in food cross-contamination [38]. According to Chepkemoi et al. [12], meat handlers who proceed to undertake non-food chores such as collecting money from clients are the most essential practices for transmitting foodborne pathogens from contaminated surfaces, resulting in food cross-contamination.

Seventeen percent (17%) and 13% of supermarket and commercial butcheries had staff workers wearing jewelry (watch, bracelet, and ring) while meat was being handled, respectively. This was a compromise in relation to the meat safety regulation R638 [26], which stipulates that wearing of jewelry, other accessories, or adornments by a person handling non-prepackaged foods or meat is strictly prohibited unless it is properly covered. This result was a lower proportion than the one reported at abattoirs and butcher shops in Bishoftu, central Ethiopia, where Bersisa et al. [7] reported that 64.5% of workers wore jewelry during working hours; a report conducted in Mekelle city, Ethiopia by Gurmu and Gebretinsae [39] showed that 66.7 percent of the workers wore jewelry materials while handling meat. Another study in Kebbi state, Nigeria by Ribah et al. [40] reported that rings, watches, and other jewelry were worn by the majority of respondents (60%) while working. Because the skin underneath jewelry provides a favorable environment for contaminating microbes, making jewelry a possible source of germs, the wearing of jewelry should be prohibited [41,42].

### 4.2. Hygiene of Working Clothes at Butcheries

In this study, among the supermarket butcheries, 67% had recent dirt (fresh particles of meat or blood) on work clothes and 17% were found with ingrained dirt (old particles of meat and blood stains) on work clothes. This indicated some level of adherence to the Department of Health [26] hygiene regulation that at all times when handling food, protective clothes must be clean and made of a material and design that cannot contaminate the food. This result differs from the report by Adzitey et al. [29] in Tamale metropolis, Ghana, in which 39% of the meat sellers appeared to be clean, with only fresh meat particles covering the vendors garments, while the rest (61%) appeared unclean with clothing ingrained with either fresh or old particles of meat. It also differs from the report by Zerabruk et al. [3] in Addis Ababa, Ethiopia, in which only 37.5% of meat handlers wore clean working clothes. According to Sulleyman et al. [17], meat contamination and subsequent food poisoning can come from poor hygiene and sanitation practices. Since the slaughter process of meat may entail a significant amount of dirty labor, working garments should be cleaned at least once every day [6].

In the present study, of all the commercial butcheries, 38% had recent dirt on their working clothes and 22% had ingrained dirt, and in all village butcheries, recent dirt covered their work clothes and 33% had ingrained dirt. This proportion is lower than that reported in Addis Ababa by Zerabruk et al. [3], in which only 37.5% of meat handlers wore clean working clothes and 62.5% wore dirty clothes. In a study by Nyamakwere et al. [6], the majority of respondents stated that they only washed their protective clothes after 3 working days. Because the butchering process may involve a lot of dirty labor, working garments should be cleaned at least once a day [6]. Personal clothes can carry microorganisms into the meat or meat-handling establishment from a range of sources. Protective clothing that has been cleaned thoroughly lessens the build-up of contamination and lowers the contamination risk. Regular cleaning is crucial to preventing contaminants from accumulating [35].

### 4.3. Infrastructure and Maintenance of Hygiene at Butcheries

During this study, all supermarket butcheries had walls, floors, and ceilings in good condition. Gutema et al. [1] reported similar results in a study conducted in slaughterhouses and retail shops in Bishoftu, Ethiopia, in which all retail shops had light bulbs, either concrete or tile floors and white painted walls and ceilings.

During the present study, every retail shop had a safe water supply. This was consistent with the demands of R 638 (Act 54 of 1972) [26], in which a sufficient supply of water must be accessible in food establishments. This result differs from the report of Adeolu et al. [43] in Karu abattoir, Abuja, Nigeria, in which more than half of the respondents (53.7%) indicated a lack of insufficient water supply and obtained their water from the tap (64.6%).

In this study, waste was not confined, properly managed or disposed of in 72% of the supermarket butcheries. This contradicted the Hygiene Requirements of R 638 [26], in which a managed waste disposal system is required in all premises. Refuse containers must be regularly cleaned and disinfected. As often as necessary, garbage must be taken out of the area where food is handled or produced, and refuse must be kept or disposed in a way which doesn’t cause a hazard. This value was higher than the one in the report of Mirembe et al. [30] in Kampala district, Uganda, wherein hardly 19 (26.0%) of the butcheries had waste storage containers and these were mostly sacks and polyethylene bags. Furthermore, in a report by Ribah et al. [40] in Kebbi state, Nigeria, 66.1% of the respondents reported inadequate waste disposal provision in major slaughter slabs of raw meat. In addition. In the report by Mbonabucha and Fweja [35] at Rungwe district, Tanzania, more than half of the butchers lacked a dustbin, which suggests improper management of the butcher waste. Wastes from abattoirs and butcheries must be disposed of carefully since they may contain pathogens that cause food-borne illnesses [44].

In this study, 13% of commercial butcheries had cracked walls and floors and 38% had dirty floors. According to Mbonabucha and Fweja [35], the cleaning process is complicated by a rough or cracked wall. A compromised physical state may compromise the effectiveness of cleansing in the butchery. Poor butchery infrastructure is one of the many major obstacles to using appropriate hygienic procedures when selling meat [19]. More focus is required on enhancing the infrastructure of slaughterhouses and retail stores as well as the government regulatory agencies’ food-quality-monitoring systems, to ensure sanitary meat production and marketing at all stages [1].

In the present study, disposable paper towels at the commercial butcheries were not available. This practice was not consistent with the requirements for premises in R 638 [26]. Yousif and Mustafa [45] reported similar results in Khartoum state, Sudan, in which all (100%) open and closed butcher shops in three regions evaluated indicated poor hygiene procedures in terms of paper towel and soap availability, Worsfold [46] also reported that staff had not been properly taught to dry equipment with paper towels, and that there was a lack of understanding of the significance of cleaning and disinfecting hand contact surfaces. The usage of soft, absorbent paper towels is more acceptable to users as it corresponds with compliance with standards recommended on hand hygiene [47].

In this study, 75% of the commercial butcheries had pest control devices. This result is similar to the report by Mbonabucha and Fweja [35] on meat protection, in which many butchers (72.1%) had mesh wire or shutter glass for the control of pests. According to the R 638 [26], food facilities must be pest-proof using the best techniques presently available, and they must have access to effective prevention measures for flies, cockroaches, and other insects.

In the present study, among the village butcheries, 50% had walls, floors, and ceilings in good condition; however, 50% had ceilings, walls that were tearing down and cracked tiles on the floor. This was not in line with R 638 [26], in which food establishments must be made of smooth, toxic-free, cleanable, non-absorbent, and dust- and water-resistant material that has no open joints on the interior surfaces of walls, ceilings, and floors. This findings were consistent with the results reported by Mirembe et al. [30] in Uganda, in which 51 (69.9%) of the butcheries were permanent structures, however inspections of the floor and walls found that 24 (32.9%) had fractured walls and 66 (90.4%) had damaged flooring. According to Yousif and Mustafa [45], the infrastructure of butcher shops is typically poor. Butcher shops are areas where there is a greater risk of meat contamination [9].

In this study, 75% of the village butcheries had a safe water supply, and 25% had water supplied from water reservoir tanks instead of directly from the tap; this water had been previously collected from the nearest borehole and stored in reservoir tanks for present and future use. This result is a lower value than the result reported by Mirembe et al. [30] in Uganda, in which there was no running water within the butcher shop and the butcheries’ major water source was tap water (91.8%), with the rest of the butcheries getting their water from neighboring protected springs. Similarly, in a report by Adeolu et al. [43], more than half of those surveyed (53.7%) reported that the water supply infrastructure was insufficient and that they get their water from the tap (64.6%). According to R 638 [26], a sufficient supply of water must be accessible in food establishments. Water used for cleaning and meat processing in butcheries and abattoirs must meet drinking water standards. Therefore, sufficient potable water must be made available to meet operational and clean-up requirements [7].

In the present study, waste was confined, managed properly, and disposed of properly in 83% of the village butcheries. This was in consistence with R 638 [26]. However, 17% of the butcheries lacked a dustbin and disposed waste on the dumpster site outside of the building of the butchery. This proportion is lower than the result of Mirembe et al. [30] in Uganda, in which hardly 19 (26.0%) of the butcheries had waste storage containers, which were mostly sacks and polythene bags, and this made it difficult to manage waste, this similar to the report by Mallhi et al. [32], in which most of the shops’ garbage was found near butchery businesses and was used to feed mice, dogs, cats, and other animals. To avoid the attraction of deadly diseases and flies, it is advised that butcher shops, similar to other food operations, be built away from damp garbage and that waste should be places durable, clean materials [35].

### 4.4. Display of Meat at Butcheries

In the present study, 83% of the supermarket butcheries had meats of different species physically separated in a meat display fridge. In comparison with the current study, Zerabruk et al. [3] reported that out of all the butcher shops, only one had a refrigerator for meat storage. Most of the beef products in the investigation were kept on hangers or tables for more than 5 h. Similarly, Ribah et al. [40] reported that fresh meat was usually sold in open markets on trays or displayed on tables without proper hygiene procedures, as well as left at room temperature. The displaying of meat in non-refrigerated conditions provides an opportunity for microbial growth [3]. The display of meat in this study in refrigerator conditions is a good practice, is in consistence with R 638 [26] and should be encouraged.

In the present study, 87% of commercial butcheries physically separated meats of different species and placed them in a window display fridge. This was consistent with the demands of R 638 [26], in which foodstuff displayed or stored must not be in direct contact with the floor, ceiling, wall, or any other surface on the ground and a chilling and freezing facility must be provided for the meat. In comparison to the present study, in the report of Smigic et al. [24], 95% of meat items were in open displays, despite the fact that butcher shops were located on the edge of the road, exposing meat to dust and smoke from passing automobiles, and just 5% were shown in a clear, closed case. Due to the fact that meat contains an abundance of nutrients required for the growth of bacteria in adequate quantities, it therefore should be stored and displayed in a refrigerator [48].

In the present study, 67% of the village butcheries had meats of different species physically separated and displayed in the refrigerator. According to Mbonabucha and Fweja [35], to ensure that meat is free of bad odors, butchers must always maintain a clean environment. The display of meat in the village butcheries is contrary to the report by Aburi [19], in which open shelter butcheries displayed meat by hanging it in the open-air most of the time, while kiosk butcheries displayed 20% of their meat on open public tables and 20% within refrigerators. This supports the view that many butcheries lack cooling facilities and therefore only stock meat that can be sold within a day [31]. Maintaining the meat’s temperature during transportation, retail display, and handling will prolong the meat’s shelf life and preserve its quality [35].

## 5. Conclusions

The present study identified good meat safety practices at supermarkets. A combination of good and unhygienic meat handling practices were identified at commercial and village butcheries. The supermarkets follow the safety procedures in the handling of meat better than all other butcheries. It is very important for other butcheries (village and commercial) to adhere to meat safety regulations. Unhygienic practices of handling meat carry the potential and high possibility of cross-contamination and may result in serious public health problems. Meat produced under unsafe handling practices may contain pathogens such as *E. coli*, *Enterobacter*, *Salmonella*, *Acinetobacter*, *Enterococci*, *Staphylococcus*, *Bacillus cereus*, *G. lamblia*, *E. histolytica* and *A. lumbricoides*, which among others may cause abdominal pains, intestinal obstruction, amoebic dysentery, liver abscesses and possible death for the Vhembe population. The study’s limitation is the lack of information on the illnesses prevalent in the Vhembe area that may be caused by these pathogenic organisms. The findings of this study suggest a need for intervention through training on food safety to improve the hygienic practices of meat handling along the beef supply chain, more especially within commercial and village butcheries.

## Figures and Tables

**Table 1 ijerph-20-02230-t001:** Survey checklist to evaluate meat safety practices and hygiene in various butcheries and supermarkets.

Checklist	Yes	No
Hygiene of Meat handlers		
Meat handlers wash hands before commencing work/prior to handling meat		
Use of gloves when meat is handled		
Hair is tied back, and hair net/cap is used		
Use of Apron/gown/coat		
Use of waterproof boots for footwear		
Protective clothes are long-sleeved and completely cover personal clothes		
Staff wears watch/jewellery while meat is handled		
Same Apron is used for different activities in the shop/butchery		
Persons handling meat also handle money		
Staff preparing and handling raw meat is separate from staff preparing and handling ready to eat meats		
Cleanliness of working clothes		
Recent dirt on working clothes		
Ingrained dirt on working clothes		
Infrastructure and maintenance of hygiene in Supermarket/butchery		
Structure of shop/butchery including walls, floors, ceilings, and fixtures are in good condition and will not yield cross contamination		
Butchery/shop floor appears clean		
Counter and hooks of butchery/shop are clean		
Cutting tables contain non-harmful materials (rust, mold)		
Disposable paper towels are available		
There is a safe water supply to the butchery/shop		
Clean equipment such as weighing scales, mincers and slicers are separately used for raw meat and ready to eat meats		
Chopping boards, knives, tongs, and other utensils are separated for raw meat and ready to eat meats		
Waste is confined, managed, and properly disposed		
Cleaning cloths and detergents are stored in sight		
Pest control devices are available		
Display of meat		
Meat of different species are physically separated and are in same window display		
Meat appears red in color and has no unpleasant odor		
Meat appears dark brown/discolored and has a strong odor		

**Table 2 ijerph-20-02230-t002:** Sociodemographic characteristics of workers at various butcheries in Vhembe district, Limpopo, South Africa.

Variables		Number (%) of Respondents (*n* = 177)
Gender	Male	94 (53)
	Female	83 (47)
Age	<25	29 (16)
	25–49	130 (73)
	50>	18 (10)
Education level	Primary education (1–8)	11 (6)
	Secondary education (9–12)	121 (68)
	Tertiary education	45 (25)
Marital status	Married	75 (42)
	Single	102 (58)
Income	R1000 & below	5 (3)
	R1001–R3000	52 (29)
	R3001–R5000	120 (68)
Location	Market/butchery area	62 (35)
	Residential area	115 (64)
Level of experience	<5 years	95(53)
	>5 years	82 (46)

**Table 3 ijerph-20-02230-t003:** Meat safety practices and hygiene at supermarkets and commercial and village butcheries in Vhembe district, Limpopo, South Africa.

Variables	Supermarkets	Commercial Butcheries	Village Butcheries
Hygiene of meat handlers			
Meat handlers wash before commencing work/prior to handling meat	Yes 13 (72%)	Yes 4 (50%)	Yes 4 (67%)
	No 5 (28%)	No 4 (50%)	No 2 (33%)
Use of gloves when meat is handled	Yes 16 (89%)	Yes 6 (75%)	Yes 1 (17%)
	No 2 (11%)	No 2 (25%)	No 5 (83%)
Hair net/cap is used	Yes 18 (100%)	Yes 6 (75%)	Yes 4 (67%)
	No 0 (0%)	No 2 (25%)	No 2 (33%)
Use of waterproof boots for footwear	Yes 18 (100%)	Yes 6 (75%)	Yes 2 (33%)
	No 0 (0%)	No 2 (25%)	No 4 (67%)
Protective clothes are long-sleeved and completely cover personal clothes	Yes 6 (33%)	Yes 3 (38%)	Yes 4 (67%)
	No 12 (67%)	No 5 (62%)	No 2 (33%)
Staff wears watch/jewelry, while meat is handled	Yes 3 (17%)	Yes 1 (13%)	Yes 2 (33%)
	No 15 (83%)	No 7 (87%)	No 4 (67%)
Same Apron is used for different activities in the shop/butchery	Yes 16 (89%)	Yes 5 (62%)	Yes 6 (100%)
	No 2 (11%)	No 3 (38%)	No 0 (0%)
Persons handling meat also handle money	Yes 0 (0%)	Yes 0 (0%)	Yes 5 (83%)
	No 18 (100%)	No 8 (100%)	No 1 (17%)
Staff preparing and handling raw meat is separate from staff preparing and handling RTE meats	Yes 16 (89%)	Yes 5 (62%)	Yes 2 (33%)
	No 2 (11%)	No 3 (38%)	No 4 (67%)
Cleanliness of working clothes			
Recent dirt on working clothes	Yes 12 (67)	Yes 3 (38)	Yes 6 (100)
	No 6 (33)	No 5 (62)	No 0 (0)
Ingrained dirt on working clothes	Yes 3 (38)	Yes 2 (25)	Yes 2 (33)
	No 15 (83)	No 6 (75)	No 4 (67)
Maintenance of hygiene and infrastructure of butchery			
Structure of shop/butchery including walls, floors, ceilings, and fixtures are in good condition and will not yield cross contamination	Yes 18 (100)	Yes 7 (87)	Yes 3 (50)
	No 0 (0)	No 1 (13)	No 3 (50)
Butchery/shop floor appears clean	Yes 7 (39)	Yes 5 (62)	Yes 5 (83)
	No 11 (61)	No 3 (38)	No 1 (17)
Counter and hooks of butchery/shop are clean	Yes 12 (67)	Yes 3 (38)	Yes 4 (67)
	No 6 (33)	No 5 (62)	No 2 (33)
Cutting tables contain non-harmful materials (rust, mold)	Yes 13 (72)	Yes 5 (62)	Yes 5 (83)
	No 5 (28)	No 3 (38)	No 1 (17)
Disposable paper towels are available	Yes 9 (50)	Yes 0 (0)	Yes 0 (0)
	No 9 (50)	No 8 (100)	No 6 (100)
There is a safe water supply to the butchery/shop	Yes 18 (100)	Yes 8 (100)	Yes 6 (100)
	No 0 (0)	No 0 (0)	No 0 (0)
Clean equipment such as weighing scales, mincers and slicers are separately used for raw meat and ready to eat meats	Yes 17 (94)	Yes 6 (75)	Yes 3 (50)
	No 1 (6)	No 2 (25)	No 3 (50)
Chopping boards, knives, tongs, and other utensils are separated for raw meat and ready to eat meats	Yes 14 (78)	Yes 5 (62)	Yes 4 (67)
	No 14 (22)	No 3 (38)	No 2 (33)
Waste is confined, managed, and properly disposed	Yes 5 (28)	Yes 6 (75)	Yes 4 (67)
	No 13 (72)	No 2 (25)	No 2 (33)
Cleaning cloths and detergents are stored in sight	Yes 15 (83)	Yes 5 (62)	Yes 4 (67)
	No 3 (17)	No 3 (38)	No 2 (33)
Pest control devices are available	Yes 18 (100)	Yes 6 (75)	Yes 2 (33)
	No 0 (0)	No 2 (25)	No 4 (67)
Display of meat			
Meat of different species are physically separated & are in the same window display	Yes 15 (83)	Yes 7 (87)	Yes 4 (67)
	No 3 (17)	No 1 (13)	No 2 (23)
Meat appears red in color & has no unpleasant odor	Yes 18 (100)	Yes 5 (62)	Yes 5 (83)
	No 0 (0)	No 3 (38)	No 1 (17)
Meat appears dark brown/discolored and has a strong odor	Yes 0 (0)	Yes 3 (38)	Yes 1 (17)
	No 18 (100)	No 5 (62)	No 5 (83)

**Table 4 ijerph-20-02230-t004:** The Mean and Standard error of meat safety practices in various butcheries and supermarkets in the Vhembe district, Limpopo, South Africa.

	Village Butcheries	Commercial Butcheries	Supermarket Butcheries
Washing hands	0.67 ± 0.21	0.50 ± 0.189	0.76 ± 0.106
Gloves	0.17 ± 0.167	0.75 ± 0.164	1.00 ± 0.000
Hairnet	0.67 ± 0.211	0.75 ± 0.164	1.00 ± 0.000
Apron	1.00 ± 0.000	1.00 ± 0.000	1.00 ± 0.000
Waterproof boots	0.33 ± 0.211	0.75 ± 0.164	1.00 ± 0.000
Protective clothes long sleeved	0.67 ± 0.211	0.50 ± 0.189	0.35 ± 0.119
Jewelry	0.33 ± 0.211	0.25 ± 0.164	0.18 ± 0.095
Same Apron is worn everywhere	1.00 ± 0.000	0.75 ± 0.164	0.88 ± 0.081
Money-handling and raw meat staff are separate	0.83 ± 0.167	0.13 ± 0.125	0.00 ± 0.000
Staff handling raw meat is similar to RTE meats	0.33 ± 0.211	0.63 ± 0.183	0.88 ± 0.081
Recent dirt on work clothes	1.00 ± 0.000	0.50 ± 0.189	0.65 ± 0.119
Ingrained dirt on work clothes	0.33 ± 0.211	0.38 ± 0.183	0.18 ± 0.095
Structure in good condition	0.50 ± 0.224	0.88 ± 0.125	1.00 ± 0.000
Floor of butchery is clean	0.83 ± 0.167	0.63 ± 0.183	0.35 ± 0.119
Counter and hooks are clean	0.67 ± 0.211	0.50 ± 0.189	0.65 ± 0.119
Cutting tables are non-harmful	0.83 ± 0.167	0.63 ± 0.183	0.76 ± 0.106
Disposable paper towels	0.00 ± 0.000	0.13 ± 0.125	0.47 ± 0.125
Safe water supply	1.00 ± 0.000	1.00 ± 0.000	1.00 ± 0.000
Weighing scales and equipment are clean	0.50 ± 0.224	0.75 ± 0.164	0.94 ± 0.059
Utensils separated for RTE and raw meats	0.67 ± 0.211	0.63 ± 0.183	0.76 ± 0.106
Waste is confined	0.67 ± 0.211	0.75 ± 0.164	0.24 ± 0.106
Pest control devices	0.33 ± 0.211	0.75 ± 0.164	1.00 ± 0.000
Clean cloths and detergents	0.67 ± 0.211	0.75 ± 0.164	0.82 ± 0.095
Meat of different species is separated	0.67 ± 0.211	0.88 ± 0.125	0.82 ± 0.095
Meat is red in color and has no unpleasant odor	0.83 ± 0.167	0.63 ± 0.183	1.00 ± 0.000
Meat is dark brown in color and has an unpleasant odor	0.17 ± 0.167	0.38 ± 0.183	0.00 ± 0.000

## Data Availability

Not applicable.

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
