# Peer review of "Evaluation of Meat Safety Practices and Hygiene among Different Butcheries and Supermarkets in Vhembe District, Limpopo Province, South Africa"

_ijerph, 2023, doi:10.3390/ijerph20032230_

Round 1
Reviewer 1 Report (Previous Reviewer 2)
I think the authors did not correct the paper as recommended.
All the findings of the paper depend on theoritical data, and the main flaw of the work stands without any comment from the authors; how did the authors justify their results, in other words, what did the authors do to confirm that the meat was without bacterial contaminations?
The authors did not do any test to confirm if the use of (for example, gloves) will reduce bacterial infections.
Author Response
Reviewer 1
Reviewers comment |
Authors response |
I think the authors did not correct the paper as recommended.
All the findings of the paper depend on theoretical data, and the main flaw of the work stands without any comment from the authors; how did the authors justify their results, in other words, what did the authors do to confirm that the meat was without bacterial contaminations? The authors did not do any test to confirm if the use of (for example, gloves) will reduce bacterial infections.
|
The remaining recommendations by the authors are now addressed as indicated and highlighted in the document.
A major issue has been brought up by the reviewer. The issue of proving bacterial contamination of meat and whether wearing gloves may lessen bacterial infections. However, it is outside the scope of the present study.
Two manuscripts will be published from our project, including this one. The first is based on the results of the currently available evaluation questionnaire, while the second is based on laboratory examination of meat samples, which will cover the prevalence and sensitivity to antimicrobials of bacterial species found in the meat. The laboratory results will be available in the second manuscript in which the authors are busy generating that data. |
Reviewer 2 Report (New Reviewer)
Lines 40-41. Check the spelling of pathogen names including capitalization (upper case) and O instead of 0 in “staphylococcus aureus, Salmonella species, Listeria monocytogenes, Escherichia coli 0157:H7 and campylobacter species” and elsewhere.
Lines 52-53. Add the countries implicated, i.e., Tanzania and Ethiopia [or Africa].
Lines 83-87. Reverse order of sentences. To protect the population from food-borne bacterial diseases, it is necessary to educate and campaign for proper sanitation and meat-handling practices in abattoirs and butcher shops [1,8,24]. However, no documentation was available with regard to meat safety practices and hygiene among butcheries and supermarkets in Vhembe district, Limpopo province, South Africa, which prompted this study.
Table 1. Modify title Cleanliness of working clothes. This is consistent with the data in Table 3.
Table 2. Under income, add currency (Rand). Is this per day, week, or year? Since 1000 Rand = about $60, this needs to be clarified, especially for readers unfamiliar with the currency used in South Africa, what do these levels mean, e.g., lower middle class, etc.
Line 159. What does a plastic mean?
Line 219. What kind of water supply? Potable running water, well water in buckets, etc.? all butcheries had a sufficient? water supply...see also lines 468-479.
Discussion. There is a certain amount of overlap between the end of the Introduction and this part of the Discussion. Check both areas to avoid duplication. “Butcheries have a huge role in prevention of meat-borne diseases because of the 224 high chances of meat contamination at butchery levels. Practice and maintenance of 225 proper hygiene during meat handling is necessary for provision of healthy and fresh 226 meat for human consumption [9].”
Lines 254-261. Butchers handling paper currency have been well-known in the past to transmit pathogens in Africa. Indicate what pathogens were present in these studies and what risks could be for the Vhembe population. Also, in lines 297-310. This is cash issue is repeated in lines 365-377, combine in one section.
Line 340. In this study, only 17% of the village butcheries wore protective gloves. This is a concern and revealed a lack…
Lines 532-533. Amplify the “unhygienic practices of handling meat to carry the potential and high possibility of cross-contamination and may result in serious public health problems” with the kinds of pathogens that could be present and link to any diseases known to occur in the community. If unavailable, this should be stated as a limitation of the study. Also, who would do the training, are there courses available, health inspector consultations?
A general comment throughout. You use South African Department of Health Regulations on General Hygiene 349 Requirements for Food Premises, Food Transport, and Related Matters R 638 (Act 54 of 350 1972) [26] frequently. Consider a shorter version after the first use, e.g., SADHR [26]. However, it should be distinguished from any other South African Department of Health Regulations if they are quoted (I didn't see any).
Author Response
Reviewer 2
Lines 40-41. Check the spelling of pathogen names including capitalization (upper case) and O instead of 0 in “staphylococcus aureus, Salmonella species, Listeria monocytogenes, Escherichia coli 0157:H7 and campylobacter species” and elsewhere.
|
The pathogen names were checked and corrected in spelling, capitalization, and italicized. Lines 40-42. |
Lines 52-53. Add the countries implicated, i.e., Tanzania and Ethiopia [or Africa].
|
The countries implicated were included. Lines 52-55. |
Lines 83-87. Reverse order of sentences. To protect the population from food-borne bacterial diseases, it is necessary to educate and campaign for proper sanitation and meat-handling practices in abattoirs and butcher shops [1,8,24]. However, no documentation was available with regard to meat safety practices and hygiene among butcheries and supermarkets in Vhembe district, Limpopo province, South Africa, which prompted this study.
|
The order of the sentences was corrected. Lines 83-88. |
Table 1. Modify title to Cleanliness of working clothes. This is consistent with the data in Table 3.
|
The title was modified.
|
Table 2. Under income, add currency (Rand). Is this per day, week, or year? Since 1000 Rand = about $60, this needs to be clarified, especially for readers unfamiliar with the currency used in South Africa, what do these levels mean, e.g., lower middle class, etc.
|
The currency is in Rand and has been stipulated for a salary of each month per employee. This means that each butchery worker belonged to the poor and lower middle class of income in South Africa. This information was added in Lines 136-138.
|
Line 159. What does a plastic mean?
|
By plastic, the authors are referring to the thin, flexible, plastic bag used for packaging foods. This statement was clarified by incorporating plastic bags in Lines 150-151 and Line 161. |
Line 219. What kind of water supply? Potable running water, well water in buckets, etc.? all butcheries had a sufficient? water supply...see also lines 468-479.
|
This statement was clarified to capture the kind of water supply being used in the village butcheries and its sufficiency. Lines 221-224 and Lines 457-460. |
Discussion. There is a certain amount of overlap between the end of the Introduction and this part of the Discussion. Check both areas to avoid duplication. “Butcheries have a huge role in prevention of meat-borne diseases because of the 224 high chances of meat contamination at butchery levels. Practice and maintenance of 225 proper hygiene during meat handling is necessary for provision of healthy and fresh 226 meat for human consumption [9].”
|
The statement was revised in the discussion. Lines 229-232. |
Lines 254-261. Butchers handling paper currency have been well-known in the past to transmit pathogens in Africa. Indicate what pathogens were present in these studies and what risks could be for the Vhembe population.
Also, in lines 297-310. This is cash issue is repeated in lines 365-377, combine in one section.
|
The pathogens in the respective previous studies have been indicated and as to what risk this could have on the Vhembe population. Lines 337-350.
The cash issue was combined in one section. Lines 319-358 |
Line 340. In this study, only 17% of the village butcheries wore protective gloves. This is a concern and revealed a lack…
|
The statement was corrected and discussed. Lines 297-300. |
Lines 532-533. Amplify the “unhygienic practices of handling meat to carry the potential and high possibility of cross-contamination and may result in serious public health problems” with the kinds of pathogens that could be present and link to any diseases known to occur in the community. If unavailable, this should be stated as a limitation of the study. Also, who would do the training, are there courses available, health inspector consultations?
|
The conclusion was amplified to include pathogens that could be present in the meat.
There is no disease known at the present to occur in the community in link to these pathogens and this was stated as the limitation of the study. Lines 523-525.
There are no courses or health inspector consultations available which would assist in training the butchery workers. We hope that with the results of this study, there Municipality department would introduce such courses in schools and training through the local health workers. |
A general comment throughout. You use the South African Department of Health Regulations on General Hygiene 349 Requirements for Food Premises, Food Transport, and Related Matters R 638 (Act 54 of 350 1972) [26] frequently. Consider a shorter version after the first use, e.g., SADHR [26]. However, it should be distinguished from any other South African Department of Health Regulations if they are quoted (I didn't see any).
|
R. 638 is recommended for use during the quotation of the regulation and has been used throughout our discussion.
|
Round 2
Reviewer 1 Report (Previous Reviewer 2)
The authors addressed to all comments
This manuscript is a resubmission of an earlier submission. The following is a list of the peer review reports and author responses from that submission.
Round 1
Reviewer 1 Report
The aim of the study was to evaluate meat safety practices and hygiene among different butcheries and retail supermarkets in Vhembe district.
Abstract
There is only a description of compliance in %. A further description of the method used is necessary to be able to analyze in greater depth what is presented. The conclusion is vague and unsupported. The objective is to evaluate meat handling practices or meat safety practices? They must be consistent
L20: This was done?
Introduction
L31: only raw meat, specify the information in a better way.
L32: change infection by illness
L48-52: it is very general. Specify how it was possible to associate these equipment and utensils with food contamination. What is used, what methods are used and how it is done.
L53-57: Only meat handlers, just reported that utensils and equipment are important. Rewrite the sentence and you can join it to the previous one.
L64-67: This sentence should go before, since it supports the problem raised.
L73: The presence of hygiene measures has an impact on hygiene, however, developed countries with excellent levels of hygiene have Foodborne illness. Review Scallan et al., 2011: 10.3201/eid1701.P11101
M&M.
L87-98: What is the hygiene standard or reference in order to determine degrees of compliance?
L100-107: Which reference was used to build this instrument. How was it validated? Was any reference used in its construction? Why was no consultation regarding age, studies, income, location to differentiate between them. They are from different butcheries and supermarkets. In addition, from different bucheries. Clarify this information.
L108: What was the construction of the instrument based on?
Results
The results are very descriptive and with little analysis in this regard. Why did you not statistically compare between items and between locations? Using only one instrument limits the results and the associated discussions.
The 3 tables can be presented in only 1 Table. Therefore, the results are scarce and not very robust.
Discussion:
L186-187: Because the authors make this claim. For example, an adequate heat treatment considerably reduces the associated risks. In addition, there is no mention that the product is consumed raw.
L197: cite correctly
L199: put period after reference 7.
The discussion is tedious since the results are scarce. 8 Pages of discussion are excessive if we consider that only one instrument was applied and also quite concise.
Conclusions.
L600-606: In the abstract it says that there were unsafe practices and here good practices. Check the wording.
Reviewer 2 Report
The current study showed some data about meat safety and hygiene practises in supermarket and butcheries. On my view the sample of this work is small (32) and the authors mentioned that they applied statistical analysis and I can not detect where these statistics. Another flaw of this manuscript is how the authors judge the applied precautions in the sites of the survey. Moreover, the authors mentioned that the objective of their work to evaluate meat safety practices and hygiene among different butcheries and retail supermarkets in Vhembe district. So, how do they evaluate this? They did only a questionnaire.